# Analysis of Cerebral Small Vessel Changes in AD Model Mice

**DOI:** 10.3390/biomedicines11010050

**Published:** 2022-12-25

**Authors:** Abu Zaffar Shibly, Abdullah Md. Sheikh, Makoto Michikawa, Shatera Tabassum, Abul Kalam Azad, Xiaojing Zhou, Yuchi Zhang, Shozo Yano, Atsushi Nagai

**Affiliations:** 1Department of Neurology, Faculty of Medicine, Shimane University, 89-1 Enya-cho, Izumo 693-8501, Japan; shibly@med.shimane-u.ac.jp (A.Z.S.); akazad88@mib.jnu.ac.bd (A.K.A.); zhouxj93@med.shimane-u.ac.jp (X.Z.); zhangyuchi1014@icloud.com (Y.Z.); 2Department of Biotechnology and Genetic Engineering, Mawlana Bhashani Science and Technology University, Santosh, Tangail 1902, Bangladesh; 3Department of Laboratory Medicine, Faculty of Medicine, Shimane University, 89-1 Enya-cho, Izumo 693-8501, Japan; abdullah@med.shimane-u.ac.jp (A.M.S.); tabassum@med.shimane-u.ac.jp (S.T.); syano@med.shimane-u.ac.jp (S.Y.); 4Department of Biochemistry, Graduate School of Medical Sciences, Nagoya City University, Nagoya 467-8601, Japan; michi@med.nagoya-cu.ac.jp

**Keywords:** Alzheimer’s disease (AD), APP transgenic mouse, amyloid β, vessel leakage, blood brain barrier (BBB)

## Abstract

Amyloid β (Aβ) peptide is deposited in the brains of sporadic Alzheimer’s disease (AD) due to impaired vessel-dependent clearance. To understand the mechanisms, we investigated time-dependent cerebrovascular changes in AD model mice. Cerebrovascular and other pathological changes were analyzed in AD model mice (J20 strain) aging from 2 to 9 months by immunostaining. At 2 months, Aβ was only intraneuronal, whereas vessels were positive from 3 months in J20 mice. Compared to wild-type (WT), vessel density was increased at 2 months but decreased at 9 months in J20 mice, claudin-5 levels were decreased, and vascular endothelial growth factor (VEGF) levels were increased in the cortex and hippocampus of J20 mice brain at all time points. Albumin extravasation was evident from 3 months in J20 brains. Collagen 4 was increased at 2 and 3 months. Aquaporin 4 was spread beyond the vessels starting from 3 months in J20, which was restricted around the vessel in wild-type mice. In conclusion, the study showed that an early decrease in claudin-5 was associated with VEGF expression, indicating dysfunction of the blood–brain barrier. Decreased claudin-5 might cause the leakage of blood constituents into the parenchyma that alters astrocyte polarity and its functions.

## 1. Introduction

Alzheimer’s disease (AD) is a common neurodegenerative disease clinically manifested as a gradual decline of memory and other cognitive functions [1,2]. The main pathological feature of the disease is the deposition of amyloid β (Aβ) peptides in the brain [1,3]. Aβ deposition is started in the neuronal populations of neocortical areas which have high metabolic activity [2]. Then from the neocortex, the deposition spreads to the allocortical areas. Aβ is produced from amyloid precursor protein (APP), a membranous protein, by β- and γ-secretase activities [4]. Numerous genetic and animal studies showed the importance of the Aβ peptide for AD pathology. For example, mutations in the APP or its processing enzymes that increase Aβ production and deposition are demonstrated to be the cause of the development of familial AD pathology [5,6]. Hence, the molecules that inhibit Aβ processing enzyme activities were considered for the therapy [7]. However, inhibiting the enzymes presented some severe adverse effects [8]. APP and its processing enzymes are evolutionary conserved, indicating that Aβ could have important roles in the biological process. Subsequently, it was demonstrated that aggregated form of Aβ is neurotoxic and neuroinflammatory [9]. Since neuroinflammation and neurodegeneration are one of the main features of AD, the process of aggregation is suggested to be important for the development of disease pathology [10]. Thus, it is suggested that in familial AD, excessive production of Aβ could overwhelm the clearance process leading to aggregation and deposition [11].

After production, Aβ is removed from the brain by enzymatic degradation [12,13,14,15], cell-mediated clearance [15,16,17,18], and clearance through perivascular pathways [19]. Tau is also deposited in the AD brain. It is reported that like Aβ, tau protein can be cleared through the perivascular pathway. In sporadic AD, no mutation was found in either APP or its processing enzymes, and the production rate of Aβ is comparable to healthy subjects. Gene association studies show a strong relationship between an apolipoprotein E, ApoE (*ApoE ε4*) variant with the disease [18,20]. Subsequently, it was found that the variant impedes Aβ clearance. In AD, Aβ is deposited in the parenchyma and around the cerebral vessels. Such findings are suggesting that disturbance in the clearance associated with the vessels could be important. Moreover, it was shown that ApoE is associated with vessel-mediated clearance of Aβ, and the efficiency of *ApoE ε4* is less than that of *ApoE ε3* [21]. Hence, sporadic AD is suggested to be caused by the disturbance in the vessel-dependent of perivascular clearance of Aβ.

The Brain lacks a lymphatic system. Instead, a perivascular pathway, so-called the glymphatic system, plays the role of the lymphatic system and is considered one of the major waste clearance systems of the brain [22,23,24,25]. The perivascular space is created initially by leptomeningeal cells surrounding the perforating arterioles (Virchow–Robin space) and contains cerebrospinal fluid (CSF). In the downstream, Virchow–Robin space is replaced by a porous basal lamina where CSF can move easily and interchange its constituents including waste products with the interstitial fluid [26]. Ultimately, the fluid in the perivascular space drains out of the brain to cervical lymph nodes. All arterioles, venules, and capillaries of the brain are surrounded by astrocytic end-feet outside of the basal lamina that creates the outer wall of the perivascular space. Importantly, a water channel protein, aquaporin 4 (AQP4), is highly expressed in astrocyte end-feet, which plays an important role in the movement of fluid in the perivascular space [27]. Aβ is demonstrated to be cleared through the perivascular space. The disturbance in the perivascular clearance due to increased load or ApoE variant could cause its deposition in the vessel wall, which interacts with the surrounding vascular cells [28]. Due to its cytotoxic [29] and inflammatory nature, such interaction could alter the vascular structure and function and affect the progression of the disease pathology.

Previous studies demonstrated that Aβ has profound effects on the cells of cerebral vessels. For example, Aβ is known to increase endothelial reactive oxygen species (ROS) production and induce endothelial dysfunction [27] and inflammatory molecule expression [30]. Moreover, the peptide can directly cause endothelial cell apoptosis. Importantly, Aβ can impair the blood–brain barrier function. All of these features can be seen in AD brains. However, there is a lack of understanding about which events appear early in AD pathology. Therefore, we have done a detailed time-dependent study using an AD model mouse to investigate the sequence of events in the pathology. We have found that at a very early age, BBB changes appear, followed by changes in vessel leakage and astrocytes’ expression of AQP4.

## 2. Materials and Methods

### 2.1. Animals

The hAPPJ20 line overexpresses the hAPP (human *APP*) containing both the Swedish (K670N/M671L) and Indiana (V717F) mutations under a platelet-derived growth factor beta-peptide (*PDGF-β*) chain promoter [31] and hemizygote males were crossed with C57Bl/6J female mice to produce hAPP-J20 (J20) and non-transgenic wild-type littermates (WT). After 1 month, all mice were housed 2–3 per cage and had access to food and water ad libitum under a 12 h light/dark cycle (lights on at 7:00 am). J20 and WT groups were identified and verified by PCR (PCR System 9700, Applied Biosystems, Waltham, MA, USA) using specific primers [32]. We used the J20 strain as an AD model and non-transgenic WT littermates (WT) as a control. Mice were fasting for 6 h before hippocampus and cortex sample collection. Male and female mice were used for all experiments. For the light microscopy studies, four age groups (2, 3, 6, and 9 months) of J20 and WT were used. Five (*n* = 5 in each group) animals were included in each four age groups. The animals used for Western blotting were decapitated, hippocampus dissected out quickly, frozen on dry ice, and stored at −80 °C. All experimental procedures and protocols were approved by the Ethical Committee of Shimane University Faculty of Medicine. (Approval number: IZ29-28). All experimental animal procedures followed the guidelines and the regulations of Experimental Animals, Shimane University, School of Medicine, Shimane, Japan and are described by the ARRIVE guidelines [33].

### 2.2. Tissue Preparation

We used WT and J20 mice for the immunohistochemical analysis. All the mice were deeply anesthetized with isoflurane and perfused transcardially with normal saline followed by 4% paraformaldehyde (PFA) in 0.1 M phosphate-buffered saline (PBS, pH-7.4). The mice’s brains were removed and post-fixed into the same fixative for 6 h to overnight. Then the brain samples were cryoprotected with 30% sucrose solution in 0.1M PBS for 48 h. Brains were serially sectioned coronally into tissue blocks of 2 mm thickness, and 10 µm thickness tissue slices were prepared as frozen tissue blocks for staining using a cryostat (Leica, Wentzler, Germany).

### 2.3. Regions of Interest (ROI)

Regions of interest were identified with reference to Pacino’s and Franklin’s mouse brain anatomy atlas [34]. For the prefrontal cortex, all six layers, between two adjacent belt transects, from the outer cortex through to the white matter border, were selected for analysis. All captured images were taken between the antero-posterior (AP) position from bregma between −1.07 mm and −2.45 mm relative to bregma for the cortical region. In addition, hippocampal CA1 regions were imaged between AP −1.67 mm and −3.51 mm relative to bregma. The reason to check the CA1 region is firstly affected hippocampal areas in the AD brain. All slides were imaged blind in random order.

### 2.4. Immunohistochemical Analysis

Quenching endogenous peroxidase activity by placing the section in 0.3% H_2_0_2_/methanol for a total of 20 min at room temperature (RT). Before quenching endogenous peroxidase activity, brain tissue sections underwent antigen retrieval based on prior antibody optimization; Single and dual immune labeling for anti-Aβ IgG underwent microwave antigen retrieval for 10–15 min in 10 mM sodium citrate buffer (pH = 6.4), Single immunolabelling for the claudin-5 underwent microwave antigen retrieval for 20 min in 10 mM sodium citrate buffer (pH = 6.4). After quenching endogenous peroxidase activity, mice brain tissue sections were incubated in a blocking reagent containing (0.1% Triton-X 100 and 5% goat, horse, and donkey serum) for 30 min at RT. The sections were incubated with primary antibody, anti-Aβ IgG (rabbit polyclonal, 1:200, Novus, Colorado Springs, CO, USA and mouse monoclonal, 1:300, Santa Cruz, CA, USA), anti-AQP4 IgG (rabbit polyclonal, 1:200, Santa Cruz, CA, USA), anti-claudin-5 IgG (rabbit polyclonal, 1:300, Abcam, Cambridge, UK), anti-albumin IgG (goat polyclonal, 1:200, Bethyl Laboratories, Inc, Montgomery, Texas, USA), anti-collagen 4 IgG (rabbit polyclonal, 1:500, Novus, Colorado Spring, CO, USA), anti-VEGF (rabbit polyclonal, 1:200, Santa Cruz, CA, USA), anti CD31 IgG (mouse monoclonal, 1:300, Santa Cruz, CA, USA), for overnight at 4 °C. Following incubation in primary antibodies, sections were incubated in biotin-conjugated anti-mouse IgG, anti-rabbit IgG, or anti-goat IgG (1:100, Vector, Ingold Road, CA, USA), followed by incubation with an avidin-biotin-peroxidase complex (ABC, Vector), for 30 min and the immune reaction products were visualized with DAB (3,3-di amino benzidine) (DAB, Sigma, St. Louis, MO, USA). The further tissue section was counterstained with Hematoxylin (Cell Path, Newtown, UK), dehydrated, and cleared in xylene before mounting in DPX (Leica, London, UK). Negative controls, with the omission of the primary antibody and isotype controls, were incubated with every run. Finally, the immunoreactive protein was visualized by reaction and analyzed with a light microscope (NIKON, ECLIPSE E600, Nikon cor., Tokyo, Japan). For immunofluorescence staining, FITC-conjugated anti-mouse, FITC-conjugated anti-goat, Texas red-conjugated anti-rabbit, or Texas red-conjugated anti-mouse antibodies were used. Stained sections were examined under a fluorescence microscope. Hoechst 33258 (Sigma, St. Louis, MO, USA) staining (10 
μg
/mL) was used to identify nuclei.

### 2.5. Image Analysis, Cell Counting, and Quantification of Immunoreactive Region of ROI

A digital microscope (NIKON, ECLIPSE 80i, Nikon cor., Tokyo, Japan) was used to capture all pictures of ROI. First, all images were imported into the NIH ImageJ analysis program version 1.52. (NIH, Bethesda, MD, USA). Then, the total cells or immunoreactive area of each microscopic field was measured by NIH ImageJ software and averaged. Next, the area and intensity threshold level analysis were determined to discriminate between immunoreactive positive staining and background labeling. Moreover, the immunoreactive area for the targeted protein was measured using interest threshold analysis with constant settings for minimum and maximum intensities. Finally, the percentage area of the positive signal was calculated by dividing the area of the positive signal by the total hippocampal or cortical area. Hoechst 33258 (Sigma, St. Louis, MO, USA) nuclear staining was used to identify the nuclei for immunofluorescence staining. All images were captured at (×40) magnification in the same plane of all experimental mice brains from three sequential microscopic fields in both hemispheres, the cortex, and the hippocampus.

### 2.6. Solanum Tuberosum Lectin (STL) Staining

STL staining was employed to visualize the brain vessel. Briefly, three tissue sections 2 mm apart were incubated with fluorescein-conjugated *Solanum tuberosum* lectin (Vector, 1:200) at room temperature for one hour. The tissue was mounted with a water-based mounting medium (Dako, Agilent, Santa Clara, CA, USA). Then a fluorescence microscope was used to examine the tissue. Photomicrographs of three microscopic fields of ROI (cortex and hippocampus) in the same plane of each hemisphere were taken at (×40) magnification. Vessel numbers were counted in a blinded manner; then averaged, the vessel number, representing the number of vessels of the mice in the designated area.

### 2.7. AQP4 Depolarization Measurement

AQP4 polarization was characterized by its dense, concentrated, and localized expression within the astrocytic end-feet, unsheathing the BBB interface of cerebral microvessels. Likewise, the loss of localized expression of AQP4 from perivascular end-feet processes and shifting toward astrocytic processes and coarse processes is referred to as depolarization or redistribution. AQP4 depolarization was measured by counting the number of vessels expressing the AQP4 beyond their perivascular astrocytic BBB interface within each field of view.

### 2.8. Western Blotting

Whole hippocampus from J20 (*n* = 3) and WT (*n* = 3) mice brains was collected at 3, 6, and 9 months of age and homogenized in 20X wt/vol of ice-cold RIPA lysis buffer (1X phosphate buffer saline, pH 7.4, 1% Nonidet p-40, 0.5% sodium deoxycholate, 0.1% SDS, 10 μg/mL PMSF, 10 μg/mL aprotinin). Under the reducing condition, 60 μg total protein of the brain tissue was separated by 10% sodium dodecyl sulfate-polyacrylamide gel electrophoresis (SDS-PAGE) and transferred to PVDF membranes (Millipore, Billerica, MA, USA). After blocking for 1 h at room temperature, the membranes were incubated with primary antibodies, anti-AQP4 IgG (1:2000, rabbit polyclonal,1:200, Santa Cruz, CA, USA), anti-claudin-5 IgG (rabbit polyclonal, 1:3000, Abcam, Cambridge, UK), anti-collagen 4 IgG (rabbit polyclonal, 1:500, Novus, Colorado Springs, CO, USA), anti-β-actin IgG (1:1000, mouse polyclonal, Santa Cruz, CA, USA) at 4 °C overnight. Subsequently, the membranes were incubated with infra-red (IR) dye-conjugated anti-rabbit or anti-mouse IgG antibodies (1:5000; LI-COR Bioscience, Lincoln, NE, USA) or a horseradish peroxidase-conjugated anti-rabbit (1:5000, Millipore Sigma, MA, USA) or anti-mouse IgG antibody (1:5000, Santa Cruz, CA, USA) for 1 h at RT for 3-, 6-, and 9-month samples. Immunoreactions were visualized using an Odyssey infrared imaging system (LI-COR Bioscience) AMERSHAM Image Quant 800 detection system (GE Healthcare, Amersham, UK). The band intensity was quantified by densitometry and the NIH ImageJ analysis software Version 1.52 (NIH, Bethesda, MD, USA). Individual expression levels of claudin-5, collagen 4, and AQP4 were normalized to the expression levels of β-actin.

### 2.9. Statistical Analysis

All numerical data are expressed as the mean ± standard deviation (SD). Statistical analysis was performed to compare mean values using Student’s *t*-test between different age-matched groups. Statistical significance was denoted as *p* < 0.05. All figures were prepared with Microsoft Excel.

## 3. Results

### 3.1. Time-Dependent Changes in the Blood Vessel Density in J20 Mouse Brains

To evaluate the blood vessel density, we stained endothelial cells with FITC-conjugated solanum tuberosum lectin (STL) (Figure 1a,b). Counting the vessels revealed that at an earlier time (2 months of age), the vessel number was higher in both cortex and hippocampus of J20 mouse brains, compared to their WT counterpart. There was no change in vessel cell density up to 6 months of age, both in the cortex and hippocampus of the J20 brain. On the other hand, at 9 months, the vessel density was decreased in both the cortex and hippocampus areas of J20 mouse brains (Figure 1c).

### 3.2. Evaluation of the Time-Dependent Changes of Aβ Deposition in J20 Mouse Brains

Next, we evaluated the time-dependent changes in Aβ deposition in J20 mouse brains. The immunostaining results showed that at 2 months of age, Aβ was mainly positive intracellularly in the neurons in both of cortex and the hippocampus. Immuno reactivity results showed no Aβ positivity in the wild type. From 3 months, Aβ was found to be deposited around vessel-like structures along with intracellular and extracellular deposition. The immunoreactivity of Aβ was also positive in the neurons and the vessels at six months of age. Extracellular Aβ deposits were increased in the time course from 6 months to 9 months of age (Figure 2a,b).

To confirm the deposition around the vessel, immunofluorescence staining of Aβ and STL staining was carried out. The staining results showed that Aβ was rarely positive around the vessels at 2 months of age both in the cortex and hippocampus of J20 mice. Only neuronal cells were positive for amyloid beta. From 3 months onwards, Aβ deposition around the vessel was observed (Figure 3a,b).

### 3.3. Time-Dependent Changes of Blood–Brain Barrier Protein Claudin-5 in J20 Mouse Brains

Since Aβ was found to be deposited around the vessels as early as 3 months of age, we investigated time-dependent changes of endothelial tight junction in J20 mouse brains. As a marker of tight junction, claudin-5 was chosen since it was reported to be altered in AD brains. Our immunostaining results showed that claudin-5 protein levels were decreased in the cortex vessels of J20 mouse brains from 3 months of age compared to their WT counterpart (Figure 4a and Appendix A). In the hippocampus, it decreased significantly from 2 months of age (Figure 4b and Appendix A). We monitored claudin-5 levels until 9 months of age and confirmed its decreased levels at every time point in the cortex and hippocampus from 3 months of age (Figure 4c,d and Appendix A). To further confirm whether the decrease in claudin-5 is independent of vessel density, we have checked the changes in claudin-5 levels per vessel by dividing the claudin-5 area by the vessel area. We found that the decrease in claudin-5 was not merely due to reduction, rather its expression was decreased in the vessels (Figure 4e,f). Moreover, our immunoblot results also showed decreased claudin-5 expression from 3 months of age in J20 mice brains (Figure 4g,h).

### 3.4. Time-Dependent Changes of Vascular Basal Lamina Protein Collagen 4 in J20 Mouse Brains

Since Aβ positive vessels were seen as early as 3 months, and the peptide is known to be cleared through the perivascular areas, we have checked the time-dependent changes in the vessel basement membrane. We chose collagen 4 as a prototype because it is known to be increased in AD brains. The immunostaining results showed anti-collagen 4 IgG stained vessel-like structures in the brains of both WT and J20 mice (Figure 5a,b). Quantification of the immunofluorescence staining revealed that at 2 and 3 months of age, collagen 4 levels were increased in both cortex and hippocampus areas of J20 mice compared to their WT counterparts (Figure 5c,d). Quantifying the immunoblot data revealed that at every age-matched time point at 3, 6, and 9 months, collagen 4 protein expression in J20 mouse brains was similar to that in WT-like littermates (Figure 5e,f).

### 3.5. Time-Dependent Extravasation of Albumin

Extravasation of albumin (70kd) is a feature of vessel leakage. We checked vessel leakage through anti-albumin IgG immunostaining. In wild-type littermate, albumin immunoreactivity was restricted within vessels at every time point in the cortex and hippocampus from 2 months of age to the end time point (9 months). At an earlier time (2 months), albumin immunoreactivity was found in the perivascular region of the vessel in the J20 mice brain (indicated by the white arrowhead) (Appendix A).

Image analysis data showed that endogenous circulating protein albumin immunoreactivity was found to diffuse outside the vessels into the parenchyma (extravasation) of J20 mice brain (indicated by white arrow) (Figure 6a,b and Appendix A). Additionally, thus extravasation events increased from 3 to 9 months of age in a time-dependent manner (Figure 6c,d and Appendix A). Furthermore, the number of % of albumin in the perivascular region of vessels increased over time in J20 mice brains (Figure 6e,f and Appendix A). Double immunofluorescence staining of an endothelial marker and albumin staining were carried out to confirm the extravasation (Appendix A).

### 3.6. Time-Dependent Changes of Water Channel Protein Aquaporin 4 in J20 Mouse Brains

In the brain, a water channel protein aquaporin 4 (AQP4) expresses exclusively in astrocyte end-feet that surround cerebral vessels, and astrocyte end-feet constitute the outer boundary of perivascular space. Moreover, AQP4 is important for perivascular fluid dynamics. Hence, we checked the AQP4 distribution in the brains of J20 mice.

The immunostaining result showed that the expressional pattern of AQP4 in the cortex and hippocampus areas of J20 mice was similar to WT at 2 months of age (Figure 7a,b). In the cortex and hippocampus of J20 mice, at three months of age, AQP4 started to spread outward from the vessels (indicated by the yellow arrowhead), which was not seen in their WT counterparts. Moreover, a grainy staining pattern was seen in the parenchyma, which was not associated with the vessel (Figure 7a,b). From 6 months onwards, some spreading of AQP4 staining was also seen in the WT cortex and hippocampus (Appendix A). However, the number of vessels showing such redistribution was significantly higher in J20 mice (Figure 7c). Furthermore, we used immunoblotting to examine the expression of AQP4. Since AQP4 redistribution begins in the brains of J20 mice at 3 months old, we examined AQP4 expression in this case from 3 to 9 months old. From 3 to 9 months, we observed no changes in the expression of AQP4 in both hippocampus of WT and J20 mice brains. (Figure 7d,e).

### 3.7. Time-Dependent Changes of VEGF Levels in J20 Mouse Brains

Claudin-5 levels were decreased in J20 mouse brains. To investigate underlying mechanisms, we checked the time-dependent changes in VEGF levels in J20 mouse brains [35]. The immunostaining results showed anti-VEGF IgG stained strongly neuronal cell and vessel-like structures in J20 mice brains at 2 and 3 months (Figure 8a,b). On the other hand, for the WT type counterpart, VEGF expression was fadely neuronal, barely vessel-like structure at 2 and 3 months of age (Figure 8a,b). The immunostaining results showed that VEGF levels were increased in J20 mice cortex and hippocampus from 2 months of age compared to their WT counterparts (Figure 8c,d and Appendix A). Such increased levels persisted at every time point that we have checked (Appendix A).

## 4. Discussion

The main finding of this study is that vascular changes start early in an APP transgenic AD mouse model rather earlier than previously detected, and the trigger could be the degradation of the blood–brain barrier proteins. Such decreased expression occurs even before the deposition of Aβ as amyloid plaques. Even immunoreactivity of Aβ was intraneuronal and around the vessels was rare at this time (2 months). Aβ is usually deposited extracellularly in the brain parenchyma as insoluble aggregates. However, soluble oligomeric aggregates were found to be the most toxic to the cells [36]. Aβ is demonstrated to be cleared from the brain through the perivascular pathway [24,28], and it is a highly aggregation-prone peptide. Hence, it is possible that during clearance, the peptide might oligomerize in the perivascular space due to its high concentration. Such oligomerized peptides could interact with surrounding endothelial cells and astrocytes and cause perivascular changes. Such changes could hamper the clearance further, which ultimately causes Aβ deposition in the brain parenchyma.

Previously it was reported that blood–brain barrier protein including claudin-5 is decreased in patients with AD [37], and in AD model animals [38]. In this study, we demonstrated that decreased claudin-5 expression is an early event. Claudin-5 expression within vessels decreased at 2 months in J20 mice brains, whereas vascular density increased at an earlier time point. So, it revealed that claudin-5 expression decreased without affecting the endothelial cell density, indicating decreased claudin-5 related to vessel leakage as at 3 months, extravasation of albumin was evident. Our claudin-5/STL expression data denote vessels themselves have decreased claudin-5 expression. During the early process of angiogenesis, tight junction protein levels decrease to allow the proliferation and migration of endothelial cells. In J20 mice brains, angiogenic growth factor VEGF levels were increased at 2 months along with increased vessel number. Such findings suggest the existence of an active angiogenic process in the brains of J20 mice that leads to decreased claudin-5 levels. Importantly, angiogenesis, especially the pathological angiogenesis process has been demonstrated to exist in the brains of AD subjects [39]. Although we did not investigate the possible mechanisms, the causes of increased VEGF levels could be the neuroinflammatory condition present in AD pathology. Aβ is known to have inflammatory properties and can induce inflammatory cytokines including IL-1β [30,40,41], and IL-1β induces VEGF expression through STAT3 and NF-κB activation [42].

VEGF was seen to be increased in J20 mice at every time point compared to WT, indicating the presence of an angiogenic signal. However, at a later time point (9 months) the vessel number was decreased in J20 mice, suggesting the coexistence of a degenerative process. Indeed, Aβ shows direct growth-inhibitory and apoptotic properties toward endothelial cells [43]. Such a degenerative process is believed to induce a hypoxic condition in AD brains and drive the angiogenesis process through the HIF-1α VEGF axis [44]. Also, the continued vascular degeneration enables the persistence of hypoxia and angiogenesis, making angiogenesis a pathological type. Vessels generated during pathological angiogenesis are usually leaky, with decreased tight junction proteins. Leaky vessels caused albumin extravasation. In our study, albumin was perivascular at 2 months and extravascular at 3 months, indicating that the BBB was impaired. As a result, blood constituents leak out into the brain parenchyma and contribute to the Aβ aggregation and deposition process. It was also reported that blood component leakage causes perivascular inflammation. This means that damage to the BBB and perivascular inflammation start the process of vascular remodeling [45]. Moreover, the perivascular space of these vessels would not be well-formed, leading to inefficient Aβ clearance.

AQP4 is mainly expressed in the astrocyte end-feet that surround the vessels [46,47]. Astrocyte end-foot forms the outer boundary of perivascular space [48]. Here, AQP4 plays an important role in perivascular fluid flow that affects the clearance of waste [49,50] and other metabolites, consequently maintaining local homeostasis. A mouse model that lacks AQP4 expression in astrocytes exhibits slowed CSF influx in perivascular space, resulting in about a 70% reduction in the clearance of interstitial solutes [24]. Inhibiting AQP4 reduces soluble Aβ drainage, implying a reduction in astrocytic end-feet contributes to Aβ deposition [51], suggesting the importance of AQP4 in AD pathology. Another important finding of this study is the redistribution of AQP4 in the brains of J20 mice. The redistribution of AQP4 in the J20 mouse brain began at 3 months of age and continued throughout the time period. Redistribution of AQP4 indicates the activation of astrocytes that participate in the neuroinflammatory process seen in AD [52]. Although the effects of AQP4 redistribution on Aβ deposition are not fully understood, a change in polarity could impede waste clearance. Taken together, such redistribution of AQP4 could start a vicious cycle by increasing the Aβ burden in brains, leading to activation of a neuroinflammatory process, which further increases the deposition. Although the cause of redistribution is not elucidated in this study, it is possible that AQP4-binding heparan sulfate proteoglycan agrin might have a role in this process [53]. Agrin is necessary for polarized distribution, and its reduction causes the redistribution of AQP4 in human glioblastoma [54]. Agrin is shown to be related to Aβ deposition, where increased agrin levels increase AQP4 [55], and consequently decrease the deposition. Hence, AQP4 redistribution might indicate reduced agrin and increased Aβ burden in J20 mice.

In this study, we demonstrated that collagen 4 levels were increased in J20 mice brains during early time points. In AD subjects, microvascular collagen contents were shown to be increased [56]. Even aging increases collagen 4 content in brain microvessels [55,57]. Since aging affects the perivascular clearance of waste, the increased collagen 4 might have a role in such disturbance of waste clearance. Although we expected to see increased collagen 4 levels in aged J20 mice, the vascular degenerative process might affect the collagen levels showing overall similar levels to WT counterparts at a later time. Collagen 4 is an important constituent of the vessel basement membrane that forms the perivascular space. Collagen 4 is shown to interact with Aβ and inhibits the fibril formation process [58]. 

However, the interaction might impede the clearance through perivascular space and cause the deposition of Aβ. We checked the changes in collagen 4 expression. Although we did not examine the basal laminal alteration through ultrastructural analysis, we need to investigate the more detailed structural changes that occurred in the early time point.

## 5. Conclusions

In conclusion, we demonstrated in this study that the blood–brain barrier functions are disturbed early in an AD mouse model which persisted throughout the investigated time period. Such disturbance of the barrier function might contribute to the perivascular changes and astrocytic neuroinflammatory process seen in AD, and establish a vicious cycle of Aβ deposition. Therefore, restoration of the blood–brain barrier functions along with increased clearance of Aβ could provide a better outcome of AD therapy.

## Figures and Tables

**Figure 1 biomedicines-11-00050-f001:**
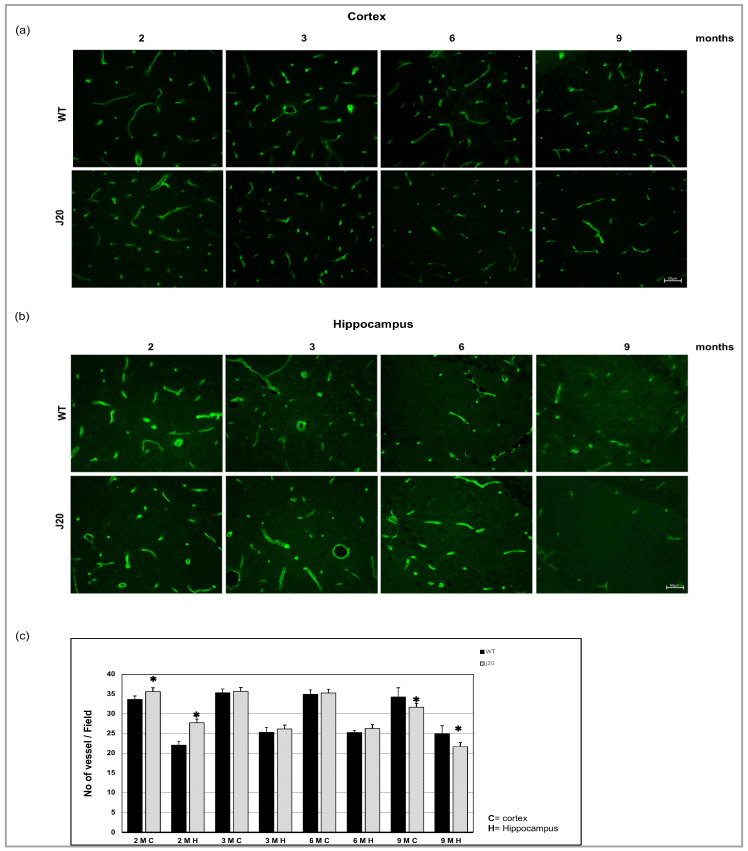
Immunofluorescence for the endothelial cell. Brain section was stained with FITC conjugated solanum tuberosum in the cortex (**a**) and hippocampus (**b**) of WT and J20 mice brain at age 2, 3, 6, and 9 months of age. (**c**) The number of vessels per field at 400 magnifications in the cortex and the hippocampal area was counted in a blinded manner. Numerical data are presented here as average ± SD, and statistical significance is denoted as * *p* < 0.05 vs. age-matched WT. M indicates month. Scale bar = 50 µm.

**Figure 2 biomedicines-11-00050-f002:**
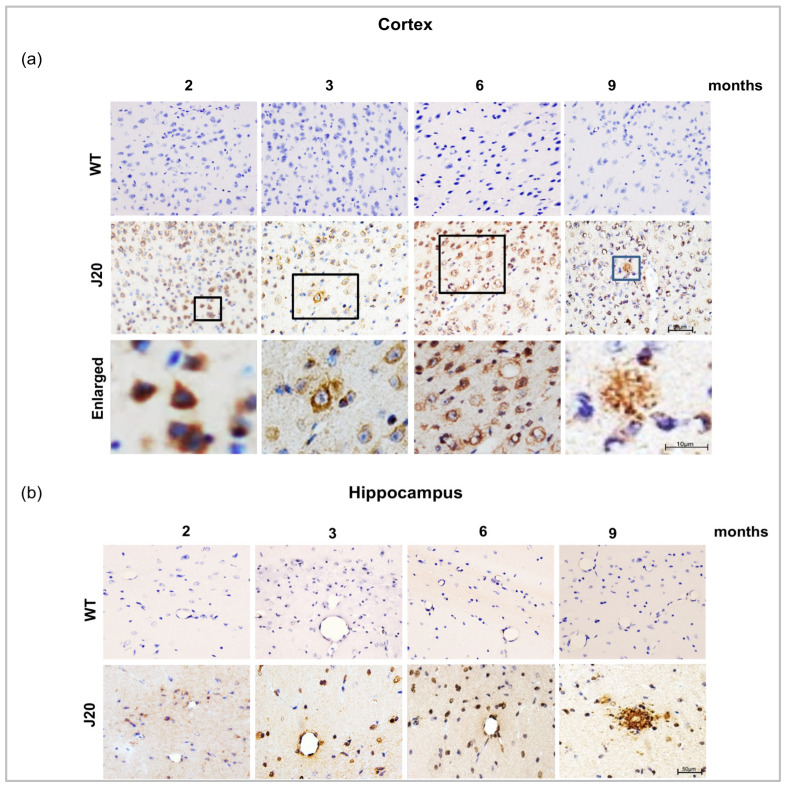
Aβ peptide immunoreactivity and vascular deposition of Aβ in J20 mice brain. (**a**,**b**) Aβ immunoreactivity in the cortex and the hippocampus of both WT and J20 mice brains at 2, 3, 6, and 9 months of age. Aβ peptides were predominantly located in the hippocampal and cortical region in J20 mice but not in WT mice brains. All boxed areas were magnified. Scale bar = 10 µm. Intraneuronal Aβ at 2 months and extravascular Aβ deposition in parenchyma at 9 months. M indicates month. Scale bar = 50 µm.

**Figure 3 biomedicines-11-00050-f003:**
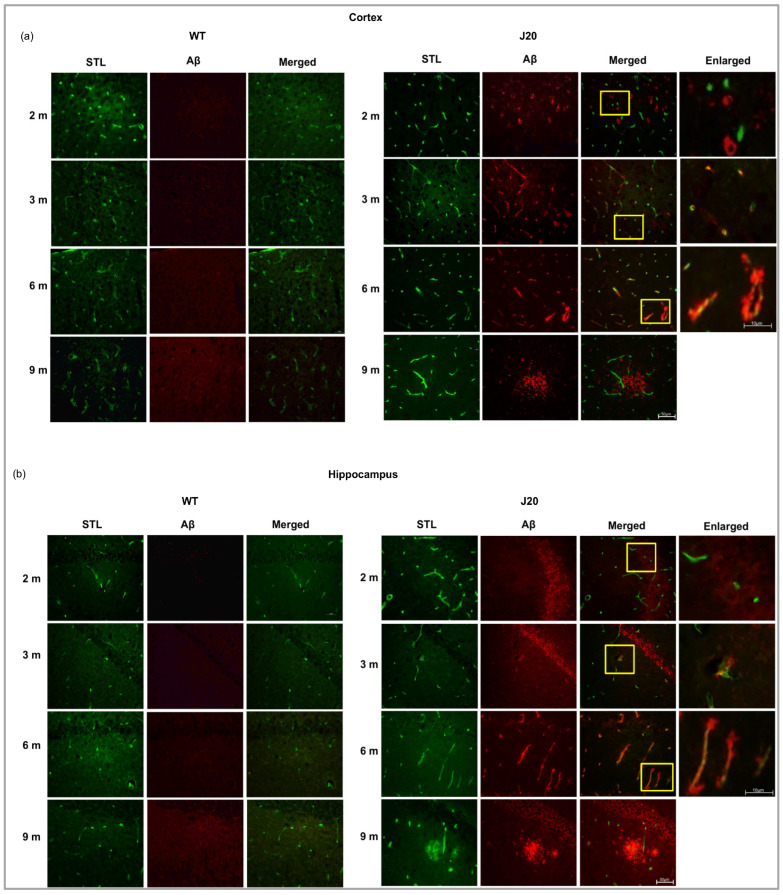
Double immunofluorescence staining of Aβ peptide and FITC conjugated solanum tuberosum in the (**a**) cortex, and the (**b**) hippocampus of both WT and J20 mice brain at age 2, 3, 6, and 9 months. Aβ immunoreactivity was positive for both neuronal and endothelial cells between 3 and 6 months, as contrasted to 2 months when Aβ peptide was solely positive for neuron cells. All yellow boxed areas were magnified. At 2 months, only neurons positive for Aβ, Aβ around the vessel at 3 months and at 6 months increased Aβ deposits around the vessel. At 9 months of age, extravascular Aβ deposition in parenchyma. M indicates month. Scale bar = 50 µm.

**Figure 4 biomedicines-11-00050-f004:**
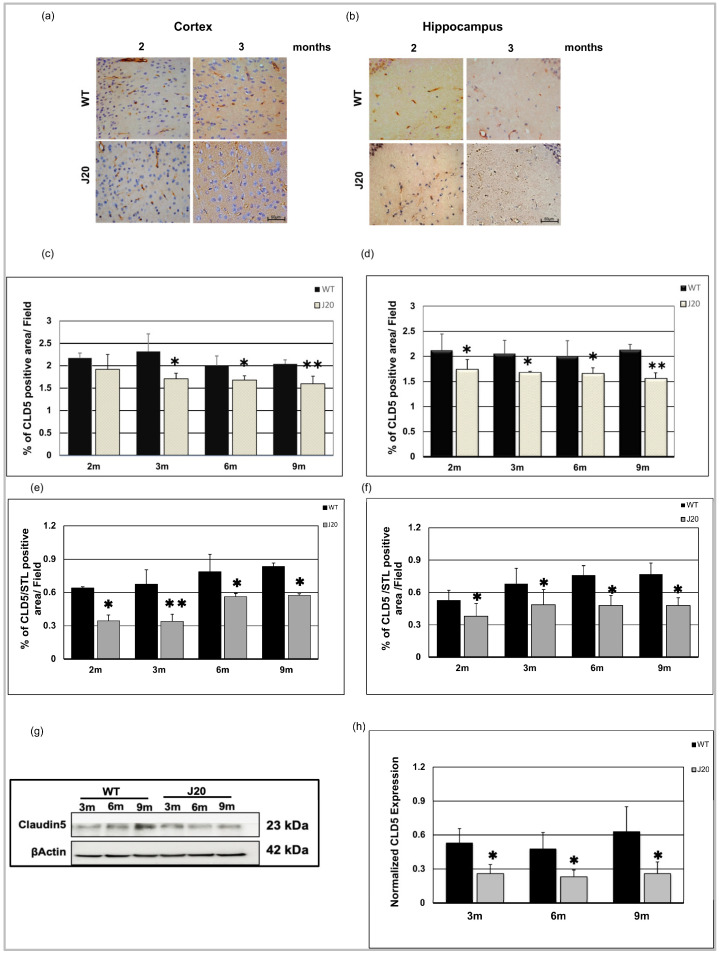
Decreased expression of claudin-5 level in J20 mice. Claudin-5 immunoreactivity in the cortex (**a**) and hippocampus (**b**) of both WT and J20 mice at 2 and 3 months. Endothelial cells of the vasculature exhibit immunoreactivity for the claudin-5 protein (brown). Decreased claudin-5 expression was found for endothelial cells in the J20 mice brain. Densitometric analysis by ImageJ as a percentage of claudin-5 positive areas per field (×40) in the cortex (**c**) and hippocampus (**d**) was shown in the graph (Appendix A). Percentage of claudin-5/STL positive areas per field (×40) in the cortex (**e**) and hippocampus (**f**) was shown in the graph. (**g**) Representative immunoblot of claudin-5 protein expression in the hippocampus of WT and J20 mice brains at 3, 6, and 9 months of age. (**h**) Normalized hippocampal claudin-5 expression of different group at 3, 6, and 9 months. Numerical data are presented here as average ± SD, and statistical significance is denoted as * *p* < 0.05, ** *p* < 0.01 vs. age-matched WT. CLD5 = claudin-5. M indicates month. Scale bar = 50 µm.

**Figure 5 biomedicines-11-00050-f005:**
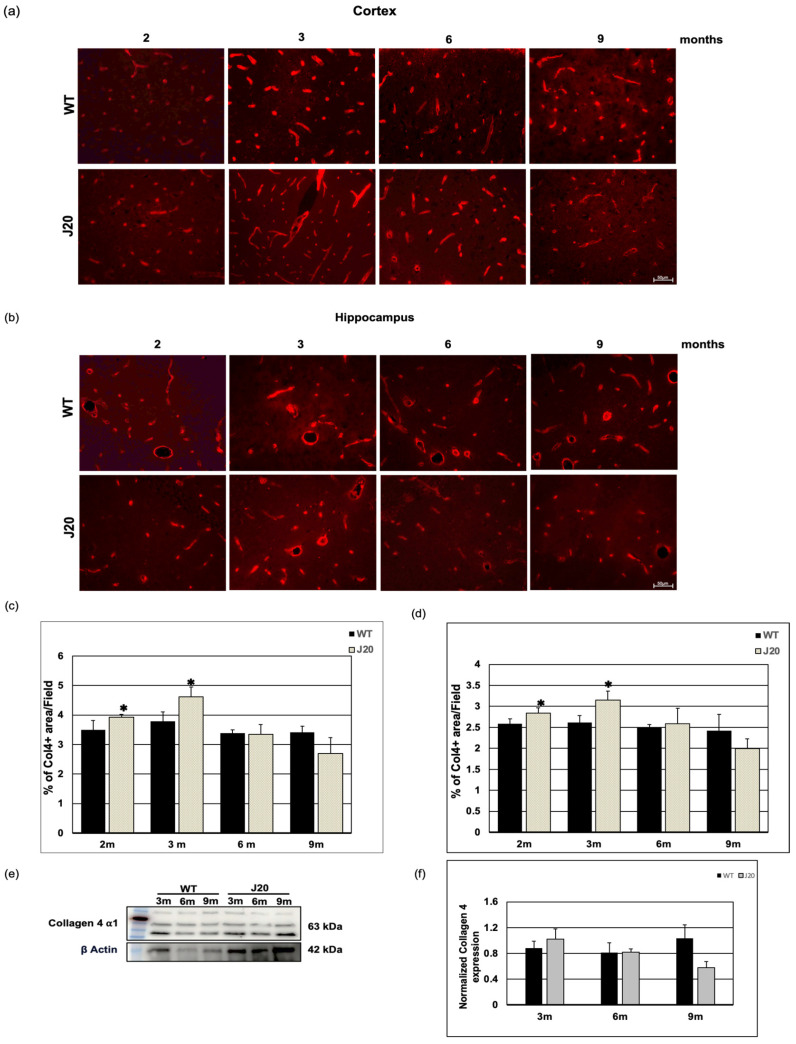
Time-dependent changes in collagen 4 levels in J20 mice brain. (**a**,**b**) Immunoreactivity of collagen 4 in the cortex (**a**) and hippocampus (**b**) of WT and J20 mice at 2, 3, 6, and 9 months of age are shown. Collagen 4 immunoreactivity was found around the vessel in WT and J20 mice brains. (**c**,**d**) Densitometric analysis was carried out using ImageJ, and the quantified data are expressed as a percent of collagen 4-positive areas per field (×40) in the cortex (**c**) and hippocampus (**d**). (**e**,**f**) A representative immunoblot of collagen 4 protein in the hippocampus of WT and J20 mice brains at 3, 6, and 9 months of age is shown here in (**e**). Quantified data of β actin normalized collagen 4 in the hippocampus at 3, 6, and 9 months of age are shown in (**f**). Numerical data are presented here as average ± SD, and statistical significance is denoted as * *p* < 0.05 vs. age-matched WT. Col4 = Collagen 4. m indicates month. Scale bar = 50 µm.

**Figure 6 biomedicines-11-00050-f006:**
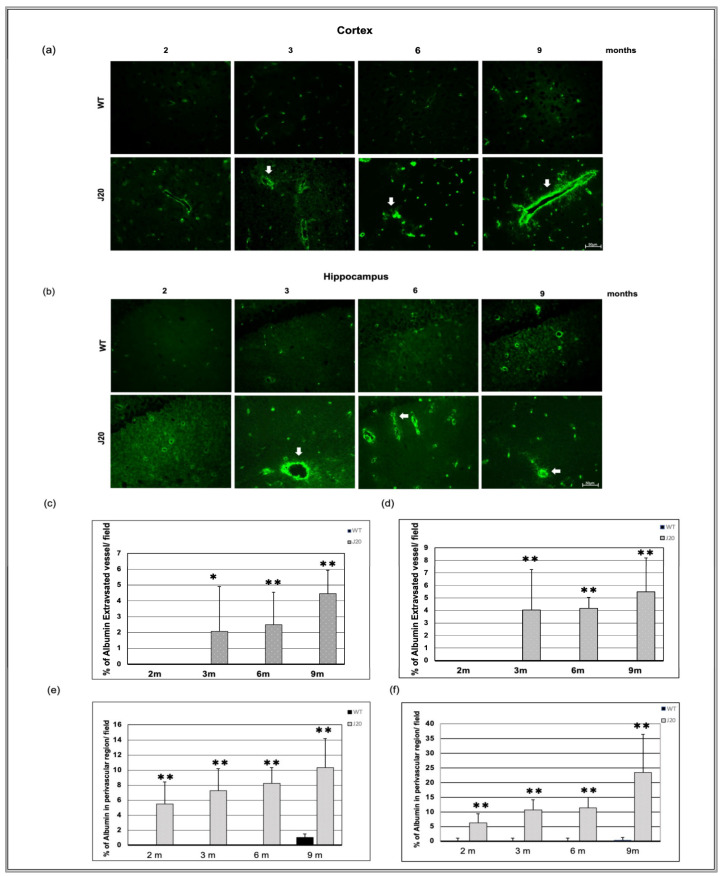
Time-dependent immunoreactivity of albumin in J20 mice brains. Albumin immunostaining in the cortex (**a**) and hippocampus (**b**) of the brains of WT and J20 mice at 2, 3, 6, and 9 months. Albumin leaked into the parenchyma (extravasation) (indicated by white arrows). The number of % of extravasated vessels per field in the cortex (**c**) and the hippocampal (**d**) area was counted at 400 magnifications in a blinded manner. Furthermore, the number of % of albumin in the perivascular region of vessels per field in the cortex (**e**) and the hippocampus (**f**) (Appendix A). Numerical data are presented here as average ± SD, and statistical significance is denoted as * *p* < 0.05, ** *p* < 0.01 vs. age-matched WT. M indicates month. Scale bar = 50 µm.

**Figure 7 biomedicines-11-00050-f007:**
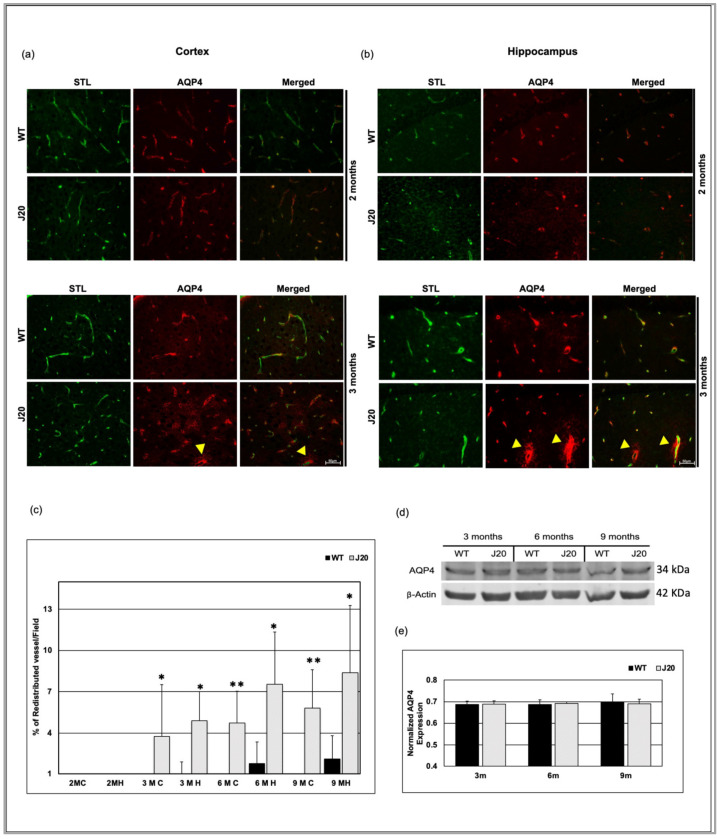
AQP4 redistribution in J20 mice brains. Immunofluorescence for the lectin, STL (green), and AQP4 (red) was shown in the cortex (**a**) and hippocampus (**b**) of WT and J20 brains at 2 and 3 months of age. At 2 months, STL/AQP4 uniform colocalization (around the vessel) was observed in the WT and J20 brain. At 3 months, intense expression of AQP4 outside the vessel in the parenchyma (redistribution) in J20 mice (indicated by yellow arrowhead). (**c**) Percentage of redistributed vessel/field in the cortex and the hippocampus of both WT and J20 mice brain at 2, 3, 6, and 9 months of age. (**d**) Representative immunoblot of AQP4 protein expression in the hippocampus of WT and J20 mice brains at 3, 6, and 9 months of age. (**e**) Normalized hippocampal AQP4 expression of different groups at 3, 6, and 9 months. Numerical data are presented here as average ± SD, and statistical significance is denoted as * *p* < 0.05, ** *p* < 0.01 vs. age-matched WT. C = cortex and H = hippocampus. M indicates month. Scale bar = 50 µm.

**Figure 8 biomedicines-11-00050-f008:**
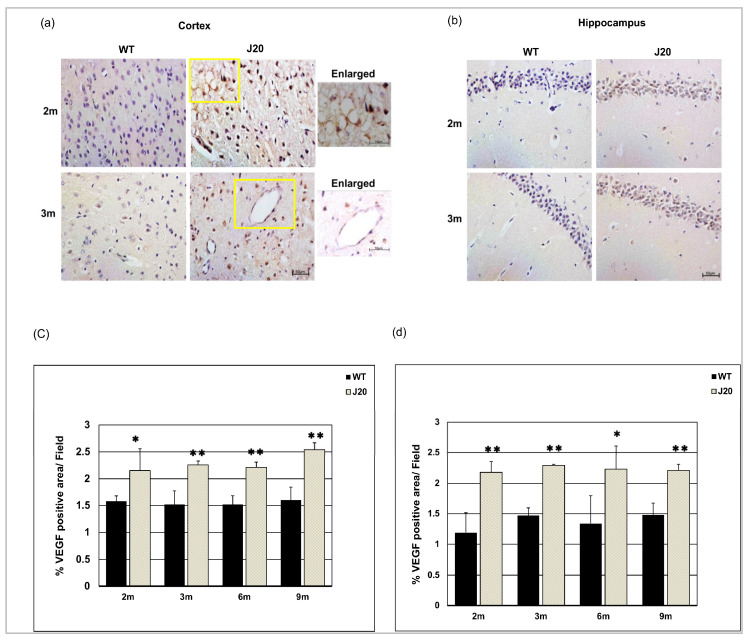
Increased expression of vascular endothelial growth factor (VEGF) levels in J20 mice brains. Immunostaining of VEGF in the cortex (**a**) and hippocampus (**b**) of both WT and J20 mice brains at 2 and 3 months. The yellow boxed areas were magnified. Scale bar = 10 µm. Densitometric analysis by ImageJ as a percentage of VEGF positive area per field (×40) in the cortex (**c**) and the hippocampus (**d**) were shown in the graph. Numerical data are presented here as average ± SD, and statistical significance is denoted as * *p* < 0.05, ** *p* < 0.01 vs. age-matched WT. M indicates the month. Scale bar = 50 µm.

## Data Availability

All data of this study are shown in the report.

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
