# Peer review of "Analysis of Cerebral Small Vessel Changes in AD Model Mice"

_biomedicines, 2022, doi:10.3390/biomedicines11010050_

Round 1

Reviewer 1 Report

Interesting study with the relevant experimental data. Methodology is properly used to address the aims of the study. Discussion can be improved by covering very recent  world- wide literature data on this topic. Also citation of the references needs to be corrected.

Author Response

Reviewer 01#

Thanks for your valuable propositions. Under your comments and recommendations on our manuscript, we tried to improve it.

Reviewer 2 Report

In the present manuscript the authors Shibly et al. investigate the cerebral microvasculature in an Alzheimers`s disease (AD) mouse model. They show age-dependent changes in vessel density and several markers of microvessels such as claudin-5, collagen 4, aquaporin 4 and VEGF levels and albumin extravasation. The authors conclude that the blood-brain barrier is disturbed early in this AD model and, thus, may initiate or contribute to neuroinflammation.

There is increasing evidence that microangiopathy is an important factor in AD, thus the authors address an important issue in their paper. However, there are several points which need to be clarified to see if the study is based on valid data and the conlusion drawn is allowed.

1.      Most parameters were investigated by microscopical analysis of brain sections immunostained for various markers. Regions of interest were cortex and hippocampus. There is no information given which layers of the cortex or which region of the hippocampus (CA1, CA3, dentate gyrus?) were investigated. There might be region-specific differences.

2.      Figure 2c,d: The authors conclude from the images that Abeta deposition can be observed around vessels in mice older than 3 months. However, this conclusion cannot be drawn from the images shown. I agree that extracellular Abeta deposition can be seen, but it is mostly not in close proximity to the vessels but appears in large clusters probably randomly distributed in the extracellular space.

3.      Figure 3: The authors show a decrease in claudin-5 positive areas of immunostained cortical and hippocampal regions in AD mice. It would be important to find out if this decreased claudin-5 staining is due to an overall decrease in vessel density (see figure 1) or if the vessels themselves have less claudin-5 expression. This is important for the conclusion that blood-brain barrier is disturbed due to claudin-5 changes.

4.      Quantification via immunostaining of brain sections: In general the superior method to quantify protein levels is western blotting.

5.      Figure 4: time-dependent changes of collagen 4 in AD mice resemble mostly the time-course of vessel density. It remains an open question if the basal lamina is altered or not. Ultrastructural analysis could help to address this issue.

Thus, the present manuscript addresses the interesting and relevant question of microvascular integrity in AD. However, the overall decrease of parameters related to endothelial integrity (claudin-5, collagen 4 and so on) does not allow the conclusion of an altered blood-brain barrier since vessel density is also decreased even it is likely that blood-brain-barrier is disturbed under these conditions. This has been shown by others in the past (see references in the manuscript) and can also be concluded from the observation of albumin extravasation.

Author Response

Reviewer 02#

Q1] Most parameters were investigated by microscopical analysis of brain sections immunostained for various markers. Regions of interest were the cortex and hippocampus. There is no information given which layers of the cortex or which region of the hippocampus (CA1, CA3, dentate gyrus?) were investigated. There might be region-specific differences.

Response: Thank you for pointing out these crucial aspects. We used a microscope to look at different markers of blood vessels in the cortex and hippocampus parts of the brain. For the prefrontal cortex, all six layers, between two adjacent belt transects, from the outer cortex through to the white matter border, were selected for analysis. All captured-images were taken between antero-posterior (AP) position from bregma between -1.07 mm to -2.45 mm relative to bregma for the cortical region. In addition, hippocampal CA1 region were imaged between AP -1.67 mm to -3.51 mm relative to bregma. I have included the information in my main manuscript. Please see on to manuscript page 3 L 126,127,128

Q2] Figure 2c, d: The authors conclude from the images that A-beta deposition can be observed around vessels in mice older than 3 months. However, this conclusion cannot be drawn from the images shown. I agree that extracellular A-beta deposition can be seen, but it is mostly not in close proximity to the vessels but appears in large clusters probably randomly distributed in the extracellular space.

Response: Under your suggestion, we tried to show the proximity in newly added photographs. Figure 2c, d is corrected per your suggestion. Figures 2c, d is moved to 3a, b. As the figure is large, we split Figure 2 in first submission into Figure 2 and 3 in the revised version. At 3 months of age, vascular Ab deposition starts, at 6 months of age, vascular Ab deposition increases. Extracellular Aβ deposits were increased in the time course from 6 months to 9 months of age. We add some lines 256,257,259,260,261.

Q3] The authors show a decrease in claudin-5 positive areas of immunostained cortical and hippocampal regions in AD mice. It would be important to find out if this decreased claudin-5 staining is due to an overall decrease in vessel density (see figure 1) or if the vessels themselves have less claudin-5 expression. This is important for the conclusion that the blood-brain barrier is disturbed due to claudin-5 changes.

Response: We agree with your suggestion. We checked the tight junction protein expression, which decreased in the endothelial cell from 2 months of age to 9 months of age in the hippocampus of the J20 mouse brain. On the other hand, the cortex was from 3 months of age. Interestingly, there was increased vessel density at two months in the J20 brain. Furthermore, there was no loss of vessel cell density up to 6 months of age, both in the hippocampus and cortex of J20 brain. As a result, it’s simple to conclude that claudin-5 expression decreased without affecting the endothelial cell density. According to your query we did CLD5/STL expression to check if the decreased claudin-5 staining is due to an overall decrease in vessel density or if the vessels themselves have less claudin-5 expression. Please see on the manuscript L 327,328,329,330.

Q4] Quantification via immunostaining of brain sections: In general, the superior method to quantify protein levels is western blotting.

Response: We share your view. We have analyzed the Claudin-5 protein via western blotting as you suggested. The expression levels of the Claudin-5 protein have been updated in Figures 4g and 4h of the manuscript. Lines 326, 327, 328, 329, 330, 337, 338, 339, and 340 are added to the original manuscript. Please see to the manuscript.

Q4] Figure 4: time-dependent changes of collagen 4 in AD mice resemble mostly the time-course of vessel density. It remains an open question if the basal lamina is altered or not. Ultrastructural analysis could help to address this issue.

Response: We agree with your suggestion. In our study, we checked of expressional changes of collagen 4 protein by Immunohistochemistry and western blotting. Immunoblot results disclosed no changes of collagen 4 expression from 3 to 9 months in J20 mice brain. We add some line on manuscript 374,375,376,377,388,389,389,390,391. Since we cannot do further ultrastructural analysis, we add this one as a limitation of our study.

Reviewer 3 Report

Alzheimer’s disease is the most common cause of dementia with no cure so far. Recently, it is suggested that impaired cerebrovascular function may precede the onset of AD. Using J20 strain AD mouse model, the authors investigated the cerebrovascular changes over 9 months.

My main concern is that the current study was done using a familial AD animal model; such a transgenic mouse model will more likely mimic the genetic form of AD and not the sporadic AD. Please see below for my point-by-point comments:

1.     In the introduction, the authors should explain briefly the clearance of tau protein by the glymphatic system.  

2.     References should be added to support the statement in page 2, lines 77, 78, and 79.

3.     In the result section, 3.1 and 3.2, please expand. 

4.     Fig 2 a and b, higher magnification images should be provided clearly showing the intracellular and extracellular Aβ deposition. Also, a control should be added to c and d. 

5.     Claudin-4 protein level should be measured using a quantitative method such as ELISA or western blot. Immunohistochemistry is a semi-quantitative method.

6.     High magnification images should be provided showing the neuronal cells and vessel-like structures in J20 mice in Fig 7. 

7.     In the abstract, line 21, replace “amyloid β” by Aβ

8.     Page 3, line 131, please correct “PH”

9.     Page 5, line 213 “he” should be “the”

10.  Page 6, line 225, add full stop after hippocampus.

11.  Add refs to support the statement in line 429 “Collagen 4 is shown to interact with Aβ and inhibits the fibril formation process”

Author Response

Reviewer 03#

Q1] In the introduction, the authors should explain briefly the clearance of tau protein by the glymphatic system. 

Response: Thank you very much for your kind suggestion. Tau is also deposited in AD and plays an important role in AD pathology. It is reported that, like Ab, Tau protein can be cleared from the brain. Aquaporin-4 (AQP4) in the glymphatic system facilitated CSF exchange with interstitial fluid (ISF), which may provide a clearance pathway for protein species such as amyloid- and tau, which accumulate in the brain in Alzheimer's disease. We explain using APP transgenic mice, the J20 mouse model, for checking Ab clearance. In J20, the Ab protein was highly expressed, with no Tau pathology. It also has some limitations. According to your suggestion, we added lines 76–77 to the manuscript.

Q2] References should be added to support the statement in page 2, lines 77, 78, and 79.

Response: Please see the main manuscript. I have included the reference into my main manuscript. Pleases see on the page 2 L 78,79

Q3] In the result section, 3.1 and 3.2, please expand. 

Response: According to your suggestion we expand result section 3.1 and 3.2 into my manuscript. Please see on the page line 222,223,256,257,259,260,261.

Q4] Fig 2 a and b, higher magnification images should be provided clearly showing the intracellular and extracellular Aβ deposition. Also, a control should be added to c and d. 

Response: According to your suggestion we rectify figure 2 a and b, and Control is added figure 2 c and d. please find figure 2c and d as figure 3a, b.

Q5] Claudin-4 protein level should be measured using a quantitative method such as ELISA or western blot. Immunohistochemistry is a semi-quantitative method.

Response: In response to your request, we have measured Claudin-5 protein levels. Figures 4e and 4f depict the expression of the Claudin-5 protein.

Q6] High magnification images should be provided showing the neuronal cells and vessel-like structures in J20 mice in Fig 7.

Response: According to your suggestion we rectify the figure 7. Please find the figure 7 as figure 8 in the manuscript. 

Q7] In the abstract, line 21, replace “amyloid β” by Aβ.

Response: We replace amyloid β by Aβ. Please see on manuscript

Q8] Page 3, line 131, please correct “PH”

Response: We apologize for the mistake. We change it. Please see it on the manuscript.

Q9] Page 5, line 213 “he” should be “the”

Response: We correct it. Please see it on the manuscript.

Q10] Page 6, line 225, add full stop after hippocampus.

Response: We correct it. Please see it on the manuscript.

Q11] Add refs to support the statement in line 429 “Collagen 4 is shown to interact with Aβ and inhibits the fibril formation process”

Response: Reference is added to the manuscript. Please find page line 545 on manuscript.

Kiuchi, Y.; Isobe, Y.; Fukushima, K. Type IV collagen prevents amyloid β-protein fibril formation. Life Sciences 2002, 70, 1555-1564.

Round 2

Reviewer 2 Report

In the revised manuscript the authors addressed all major points and the quality of the paper increased significantly.

Author Response

Reviewer 02#

Response:

We are grateful to you for your valuable comments and recommendations.

We have thoroughly checked the spells in our manuscript, corrected some mistakes, and improved it also under the guest editor’s suggestions. Now, our manuscript is ready for waiting for your consideration.

Reviewer 3 Report

The authors have addressed my comments/suggestions

Author Response

Reviewer 03#

Response:

We are grateful to you for your valuable comments and recommendations.

We have thoroughly checked the spells in our manuscript, correct some mistakes and improved it also under the guest editor’s suggestions. Now, our manuscript is ready for waiting for your consideration.